# Faster R-CNN: Towards Real-Time Object Detection with Region Proposal Networks

**Shaoqing Ren**[*]   **Kaiming He**   **Ross Girshick**   **Jian Sun**

Microsoft Research

{v-shren, kahe, rbg, jiansun}@microsoft.com

## Abstract

State-of-the-art object detection networks depend on region proposal algorithms to hypothesize object locations. Advances like SPPnet [7] and Fast R-CNN [5] have reduced the running time of these detection networks, exposing region proposal computation as a bottleneck. In this work, we introduce a *Region Proposal Network* (RPN) that shares full-image convolutional features with the detection network, thus enabling nearly cost-free region proposals. An RPN is a fully-convolutional network that simultaneously predicts object bounds and objectness scores at each position. RPNs are trained end-to-end to generate high-quality region proposals, which are used by Fast R-CNN for detection. With a simple alternating optimization, RPN and Fast R-CNN can be trained to share convolutional features. For the very deep VGG-16 model [19], our detection system has a frame rate of 5fps (*including all steps*) on a GPU, while achieving state-of-the-art object detection accuracy on PASCAL VOC 2007 (73.2% mAP) and 2012 (70.4% mAP) using 300 proposals per image. Code is available at https://github.com/ShaoqingRen/faster_rcnn.

## 1   Introduction

Recent advances in object detection are driven by the success of region proposal methods (*e.g.*, [22]) and region-based convolutional neural networks (R-CNNs) [6]. Although region-based CNNs were computationally expensive as originally developed in [6], their cost has been drastically reduced thanks to sharing convolutions across proposals [7, 5]. The latest incarnation, Fast R-CNN [5], achieves near real-time rates using very deep networks [19], *when ignoring the time spent on region proposals*. Now, proposals are the computational bottleneck in state-of-the-art detection systems.

Region proposal methods typically rely on inexpensive features and economical inference schemes. Selective Search (SS) [22], one of the most popular methods, greedily merges superpixels based on engineered low-level features. Yet when compared to efficient detection networks [5], Selective Search is an order of magnitude slower, at 2s per image in a CPU implementation. EdgeBoxes [24] currently provides the best tradeoff between proposal quality and speed, at 0.2s per image. Nevertheless, the region proposal step still consumes as much running time as the detection network.

One may note that fast region-based CNNs take advantage of GPUs, while the region proposal methods used in research are implemented on the CPU, making such runtime comparisons inequitable. An obvious way to accelerate proposal computation is to re-implement it for the GPU. This may be an effective engineering solution, but re-implementation ignores the down-stream detection network and therefore misses important opportunities for sharing computation.

In this paper, we show that an algorithmic change—computing proposals with a deep net—leads to an elegant and effective solution, where proposal computation is nearly cost-free given the de-

---

[*]Shaoqing Ren is with the University of Science and Technology of China. This work was done when he was an intern at Microsoft Research.

tection network's computation. To this end, we introduce novel *Region Proposal Networks* (RPNs) that share convolutional layers with state-of-the-art object detection networks [7, 5]. By sharing convolutions at test-time, the marginal cost for computing proposals is small (*e.g.*, 10ms per image).

Our observation is that the convolutional (conv) feature maps used by region-based detectors, like Fast R-CNN, can also be used for generating region proposals. On top of these conv features, we construct RPNs by adding two additional conv layers: one that encodes each conv map position into a short (*e.g.*, 256-d) feature vector and a second that, at each conv map position, outputs an objectness score and regressed bounds for $k$ region proposals relative to various scales and aspect ratios at that location ($k = 9$ is a typical value).

Our RPNs are thus a kind of fully-convolutional network (FCN) [14] and they can be trained end-to-end specifically for the task for generating detection proposals. To unify RPNs with Fast R-CNN [5] object detection networks, we propose a simple training scheme that alternates between fine-tuning for the region proposal task and then fine-tuning for object detection, while keeping the proposals fixed. This scheme converges quickly and produces a unified network with conv features that are shared between both tasks.

We evaluate our method on the PASCAL VOC detection benchmarks [4], where RPNs with Fast R-CNNs produce detection accuracy better than the strong baseline of Selective Search with Fast R-CNNs. Meanwhile, our method waives nearly all computational burdens of SS at test-time—the effective running time for proposals is just 10 milliseconds. Using the expensive very deep models of [19], our detection method still has a frame rate of 5fps (*including all steps*) on a GPU, and thus is a practical object detection system in terms of both speed and accuracy (73.2% mAP on PASCAL VOC 2007 and 70.4% mAP on 2012). Code is available at `https://github.com/ShaoqingRen/faster_rcnn`.

## 2   Related Work

Several recent papers have proposed ways of using deep networks for locating class-specific or class-agnostic bounding boxes [21, 18, 3, 20]. In the OverFeat method [18], a fully-connected (fc) layer is trained to predict the box coordinates for the localization task that assumes a single object. The fc layer is then turned into a conv layer for detecting multiple class-specific objects. The Multi-Box methods [3, 20] generate region proposals from a network whose last fc layer simultaneously predicts multiple (*e.g.*, 800) boxes, which are used for R-CNN [6] object detection. Their proposal network is applied on a single image or multiple large image crops (*e.g.*, 224×224) [20]. We discuss OverFeat and MultiBox in more depth later in context with our method.

Shared computation of convolutions [18, 7, 2, 5] has been attracting increasing attention for efficient, yet accurate, visual recognition. The OverFeat paper [18] computes conv features from an image pyramid for classification, localization, and detection. Adaptively-sized pooling (SPP) [7] on shared conv feature maps is proposed for efficient region-based object detection [7, 16] and semantic segmentation [2]. Fast R-CNN [5] enables end-to-end detector training on shared conv features and shows compelling accuracy and speed.

## 3   Region Proposal Networks

A Region Proposal Network (RPN) takes an image (of any size) as input and outputs a set of rectangular object proposals, each with an objectness score.[1] We model this process with a fully-convolutional network [14], which we describe in this section. Because our ultimate goal is to share computation with a Fast R-CNN object detection network [5], we assume that both nets share a common set of conv layers. In our experiments, we investigate the Zeiler and Fergus model [23] (ZF), which has 5 shareable conv layers and the Simonyan and Zisserman model [19] (VGG), which has 13 shareable conv layers.

To generate region proposals, we slide a small network over the conv feature map output by the last shared conv layer. This network is fully connected to an $n \times n$ spatial window of the input conv

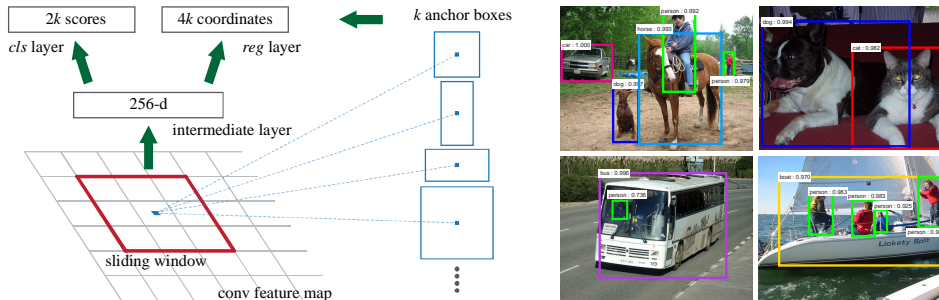

Figure 1: **Left**: Region Proposal Network (RPN). **Right**: Example detections using RPN proposals on PASCAL VOC 2007 test. Our method detects objects in a wide range of scales and aspect ratios.

feature map. Each sliding window is mapped to a lower-dimensional vector (256-d for ZF and 512-d for VGG). This vector is fed into two sibling fully-connected layers—a box-regression layer (*reg*) and a box-classification layer (*cls*). We use $n = 3$ in this paper, noting that the effective receptive field on the input image is large (171 and 228 pixels for ZF and VGG, respectively). This mini-network is illustrated at a single position in Fig. 1 (left). Note that because the mini-network operates in a sliding-window fashion, the fully-connected layers are shared across all spatial locations. This architecture is naturally implemented with an $n \times n$ conv layer followed by two sibling $1 \times 1$ conv layers (for *reg* and *cls*, respectively). ReLUs [15] are applied to the output of the $n \times n$ conv layer.

**Translation-Invariant Anchors**

At each sliding-window location, we simultaneously predict $k$ region proposals, so the *reg* layer has $4k$ outputs encoding the coordinates of $k$ boxes. The *cls* layer outputs $2k$ scores that estimate probability of object / not-object for each proposal.[2] The $k$ proposals are parameterized *relative* to $k$ reference boxes, called *anchors*. Each anchor is centered at the sliding window in question, and is associated with a scale and aspect ratio. We use 3 scales and 3 aspect ratios, yielding $k = 9$ anchors at each sliding position. For a conv feature map of a size $W \times H$ (typically ~2,400), there are $WHk$ anchors in total. An important property of our approach is that it is *translation invariant*, both in terms of the anchors and the functions that compute proposals relative to the anchors.

As a comparison, the MultiBox method [20] uses k-means to generate 800 anchors, which are *not* translation invariant. If one translates an object in an image, the proposal should translate and the same function should be able to predict the proposal in either location. Moreover, because the MultiBox anchors are not translation invariant, it requires a $(4+1)\times800$-dimensional output layer, whereas our method requires a $(4+2)\times9$-dimensional output layer. Our proposal layers have an order of magnitude fewer parameters (27 million for MultiBox using GoogLeNet [20] *vs.* 2.4 million for RPN using VGG-16), and thus have less risk of overfitting on small datasets, like PASCAL VOC.

**A Loss Function for Learning Region Proposals**

For training RPNs, we assign a binary class label (of being an object or not) to each anchor. We assign a positive label to two kinds of anchors: (i) the anchor/anchors with the highest Intersection-over-Union (IoU) overlap with a ground-truth box, or (ii) an anchor that has an IoU overlap higher than 0.7 with any ground-truth box. Note that a single ground-truth box may assign positive labels to multiple anchors. We assign a negative label to a non-positive anchor if its IoU ratio is lower than 0.3 for all ground-truth boxes. Anchors that are neither positive nor negative do not contribute to the training objective.

With these definitions, we minimize an objective function following the multi-task loss in Fast R-CNN [5]. Our loss function for an image is defined as:

$$L(\{p_i\}, \{t_i\}) = \frac{1}{N_{cls}} \sum_i L_{cls}(p_i, p_i^*) + \lambda \frac{1}{N_{reg}} \sum_i p_i^* L_{reg}(t_i, t_i^*). \tag{1}$$

Here, $i$ is the index of an anchor in a mini-batch and $p_i$ is the predicted probability of anchor $i$ being an object. The ground-truth label $p_i^*$ is 1 if the anchor is positive, and is 0 if the anchor is negative. $t_i$ is a vector representing the 4 parameterized coordinates of the predicted bounding box, and $t_i^*$ is that of the ground-truth box associated with a positive anchor. The classification loss $L_{cls}$ is log loss over two classes (object *vs.* not object). For the regression loss, we use $L_{reg}(t_i, t_i^*) = R(t_i - t_i^*)$ where $R$ is the robust loss function (smooth $L_1$) defined in [5]. The term $p_i^* L_{reg}$ means the regression loss is activated only for positive anchors ($p_i^* = 1$) and is disabled otherwise ($p_i^* = 0$). The outputs of the *cls* and *reg* layers consist of $\{p_i\}$ and $\{t_i\}$ respectively. The two terms are normalized with $N_{cls}$ and $N_{reg}$, and a balancing weight $\lambda$.[3]

For regression, we adopt the parameterizations of the 4 coordinates following [6]:

$$t_{\mathrm{x}} = (x - x_{\mathrm{a}})/w_{\mathrm{a}}, \quad t_{\mathrm{y}} = (y - y_{\mathrm{a}})/h_{\mathrm{a}}, \quad t_{\mathrm{w}} = \log(w/w_{\mathrm{a}}), \quad t_{\mathrm{h}} = \log(h/h_{\mathrm{a}}),$$
$$t_{\mathrm{x}}^* = (x^* - x_{\mathrm{a}})/w_{\mathrm{a}}, \quad t_{\mathrm{y}}^* = (y^* - y_{\mathrm{a}})/h_{\mathrm{a}}, \quad t_{\mathrm{w}}^* = \log(w^*/w_{\mathrm{a}}), \quad t_{\mathrm{h}}^* = \log(h^*/h_{\mathrm{a}}),$$

where $x$, $y$, $w$, and $h$ denote the two coordinates of the box center, width, and height. Variables $x$, $x_{\mathrm{a}}$, and $x^*$ are for the predicted box, anchor box, and ground-truth box respectively (likewise for $y, w, h$). This can be thought of as bounding-box regression from an anchor box to a nearby ground-truth box.

Nevertheless, our method achieves bounding-box regression by a different manner from previous feature-map-based methods [7, 5]. In [7, 5], bounding-box regression is performed on features pooled from *arbitrarily* sized regions, and the regression weights are *shared* by all region sizes. In our formulation, the features used for regression are of the *same* spatial size ($n \times n$) on the feature maps. To account for varying sizes, a set of $k$ bounding-box regressors are learned. Each regressor is responsible for one scale and one aspect ratio, and the $k$ regressors do *not* share weights. As such, it is still possible to predict boxes of various sizes even though the features are of a fixed size/scale.

**Optimization**

The RPN, which is naturally implemented as a fully-convolutional network [14], can be trained end-to-end by back-propagation and stochastic gradient descent (SGD) [12]. We follow the "image-centric" sampling strategy from [5] to train this network. Each mini-batch arises from a single image that contains many positive and negative anchors. It is possible to optimize for the loss functions of all anchors, but this will bias towards negative samples as they are dominate. Instead, we randomly sample 256 anchors in an image to compute the loss function of a mini-batch, where the sampled positive and negative anchors have a ratio of *up to* 1:1. If there are fewer than 128 positive samples in an image, we pad the mini-batch with negative ones.

We randomly initialize all new layers by drawing weights from a zero-mean Gaussian distribution with standard deviation 0.01. All other layers (*i.e.*, the shared conv layers) are initialized by pre-training a model for ImageNet classification [17], as is standard practice [6]. We tune all layers of the ZF net, and conv3_1 and up for the VGG net to conserve memory [5]. We use a learning rate of 0.001 for 60k mini-batches, and 0.0001 for the next 20k mini-batches on the PASCAL dataset. We also use a momentum of 0.9 and a weight decay of 0.0005 [11]. Our implementation uses Caffe [10].

**Sharing Convolutional Features for Region Proposal and Object Detection**

Thus far we have described how to train a network for region proposal generation, without considering the region-based object detection CNN that will utilize these proposals. For the detection network, we adopt Fast R-CNN [5][4] and now describe an algorithm that learns conv layers that are shared between the RPN and Fast R-CNN.

Both RPN and Fast R-CNN, trained independently, will modify their conv layers in different ways. We therefore need to develop a technique that allows for sharing conv layers between the two networks, rather than learning two separate networks. Note that this is not as easy as simply defining a single network that includes both RPN and Fast R-CNN, and then optimizing it jointly with back-propagation. The reason is that Fast R-CNN training depends on *fixed* object proposals and it is

not clear a priori if learning Fast R-CNN while simultaneously changing the proposal mechanism will converge. While this joint optimizing is an interesting question for future work, we develop a pragmatic 4-step training algorithm to learn shared features via alternating optimization.

In the first step, we train the RPN as described above. This network is initialized with an ImageNet-pre-trained model and fine-tuned end-to-end for the region proposal task. In the second step, we train a separate detection network by Fast R-CNN using the proposals generated by the step-1 RPN. This detection network is also initialized by the ImageNet-pre-trained model. At this point the two networks do not share conv layers. In the third step, we use the detector network to initialize RPN training, but we fix the shared conv layers and only fine-tune the layers unique to RPN. Now the two networks share conv layers. Finally, keeping the shared conv layers fixed, we fine-tune the fc layers of the Fast R-CNN. As such, both networks share the same conv layers and form a unified network.

**Implementation Details**

We train and test both region proposal and object detection networks on single-scale images [7, 5]. We re-scale the images such that their shorter side is $s = 600$ pixels [5]. Multi-scale feature extraction may improve accuracy but does not exhibit a good speed-accuracy trade-off [5]. We also note that for ZF and VGG nets, the total stride on the last conv layer is 16 pixels on the re-scaled image, and thus is $\sim$10 pixels on a typical PASCAL image ($\sim$500$\times$375). Even such a large stride provides good results, though accuracy may be further improved with a smaller stride.

For anchors, we use 3 scales with box areas of $128^2$, $256^2$, and $512^2$ pixels, and 3 aspect ratios of 1:1, 1:2, and 2:1. We note that our algorithm allows the use of anchor boxes that are larger than the underlying receptive field when predicting large proposals. Such predictions are not impossible—one may still roughly infer the extent of an object if only the middle of the object is visible. With this design, our solution does not need multi-scale features or multi-scale sliding windows to predict large regions, saving considerable running time. Fig. 1 (right) shows the capability of our method for a wide range of scales and aspect ratios. The table below shows the *learned average proposal size* for each anchor using the ZF net (numbers for $s = 600$).

| anchor | $128^2$, 2:1 | $128^2$, 1:1 | $128^2$, 1:2 | $256^2$, 2:1 | $256^2$, 1:1 | $256^2$, 1:2 | $512^2$, 2:1 | $512^2$, 1:1 | $512^2$, 1:2 |
|---|---|---|---|---|---|---|---|---|---|
| proposal | 188$\times$111 | 113$\times$114 | 70$\times$92 | 416$\times$229 | 261$\times$284 | 174$\times$332 | 768$\times$437 | 499$\times$501 | 355$\times$715 |

The anchor boxes that cross image boundaries need to be handled with care. During training, we ignore all cross-boundary anchors so they do not contribute to the loss. For a typical $1000 \times 600$ image, there will be roughly 20k ($\approx 60 \times 40 \times 9$) anchors in total. With the cross-boundary anchors ignored, there are about 6k anchors per image for training. If the boundary-crossing outliers are not ignored in training, they introduce large, difficult to correct error terms in the objective, and training does not converge. During testing, however, we still apply the fully-convolutional RPN to the entire image. This may generate cross-boundary proposal boxes, which we clip to the image boundary.

Some RPN proposals highly overlap with each other. To reduce redundancy, we adopt non-maximum suppression (NMS) on the proposal regions based on their *cls* scores. We fix the IoU threshold for NMS at 0.7, which leaves us about 2k proposal regions per image. As we will show, NMS does not harm the ultimate detection accuracy, but substantially reduces the number of proposals. After NMS, we use the top-$N$ ranked proposal regions for detection. In the following, we train Fast R-CNN using 2k RPN proposals, but evaluate different numbers of proposals at test-time.

## 4 Experiments

We comprehensively evaluate our method on the PASCAL VOC 2007 detection benchmark [4]. This dataset consists of about 5k trainval images and 5k test images over 20 object categories. We also provide results in the PASCAL VOC 2012 benchmark for a few models. For the ImageNet pre-trained network, we use the "fast" version of ZF net [23] that has 5 conv layers and 3 fc layers, and the public VGG-16 model[5] [19] that has 13 conv layers and 3 fc layers. We primarily evaluate detection mean Average Precision (mAP), because this is the actual metric for object detection (rather than focusing on object proposal proxy metrics).

Table 1 (top) shows Fast R-CNN results when trained and tested using various region proposal methods. These results use the ZF net. For Selective Search (SS) [22], we generate about 2k SS

Table 1: Detection results on **PASCAL VOC 2007 test set** (trained on VOC 2007 trainval). The detectors are Fast R-CNN with ZF, but using various proposal methods for training and testing.

| train-time region proposals | | test-time region proposals | | |
| method | # boxes | method | # proposals | mAP (%) |
| --- | --- | --- | --- | --- |
| SS | 2k | SS | 2k | 58.7 |
| EB | 2k | EB | 2k | 58.6 |
| RPN+ZF, shared | 2k | RPN+ZF, shared | 300 | **59.9** |
| *ablation experiments follow below* | | | | |
| RPN+ZF, unshared | 2k | RPN+ZF, unshared | 300 | 58.7 |
| SS | 2k | RPN+ZF | 100 | 55.1 |
| SS | 2k | RPN+ZF | 300 | 56.8 |
| SS | 2k | RPN+ZF | 1k | 56.3 |
| SS | 2k | RPN+ZF (no NMS) | 6k | 55.2 |
| SS | 2k | RPN+ZF (no *cls*) | 100 | 44.6 |
| SS | 2k | RPN+ZF (no *cls*) | 300 | 51.4 |
| SS | 2k | RPN+ZF (no *cls*) | 1k | 55.8 |
| SS | 2k | RPN+ZF (no *reg*) | 300 | 52.1 |
| SS | 2k | RPN+ZF (no *reg*) | 1k | 51.3 |
| SS | 2k | RPN+VGG | 300 | 59.2 |

proposals by the "fast" mode. For EdgeBoxes (EB) [24], we generate the proposals by the default EB setting tuned for 0.7 IoU. SS has an mAP of 58.7% and EB has an mAP of 58.6%. RPN with Fast R-CNN achieves competitive results, with an mAP of 59.9% while using *up to* 300 proposals[6]. Using RPN yields a much faster detection system than using either SS or EB because of shared conv computations; the fewer proposals also reduce the region-wise fc cost. Next, we consider several ablations of RPN and then show that proposal quality improves when using the very deep network.

**Ablation Experiments.** To investigate the behavior of RPNs as a proposal method, we conducted several ablation studies. First, we show the effect of sharing conv layers between the RPN and Fast R-CNN detection network. To do this, we stop after the second step in the 4-step training process. Using separate networks reduces the result slightly to 58.7% (RPN+ZF, unshared, Table 1). We observe that this is because in the third step when the detector-tuned features are used to fine-tune the RPN, the proposal quality is improved.

Next, we disentangle the RPN's influence on training the Fast R-CNN detection network. For this purpose, we train a Fast R-CNN model by using the 2k SS proposals and ZF net. We fix this detector and evaluate the detection mAP by changing the proposal regions used at test-time. In these ablation experiments, the RPN does not share features with the detector.

Replacing SS with 300 RPN proposals at test-time leads to an mAP of 56.8%. The loss in mAP is because of the inconsistency between the training/testing proposals. This result serves as the baseline for the following comparisons.

Somewhat surprisingly, the RPN still leads to a competitive result (55.1%) when using the top-ranked 100 proposals at test-time, indicating that the top-ranked RPN proposals are accurate. On the other extreme, using the top-ranked 6k RPN proposals (without NMS) has a comparable mAP (55.2%), suggesting NMS does not harm the detection mAP and may reduce false alarms.

Next, we separately investigate the roles of RPN's *cls* and *reg* outputs by turning off either of them at test-time. When the *cls* layer is removed at test-time (thus no NMS/ranking is used), we randomly sample $N$ proposals from the unscored regions. The mAP is nearly unchanged with $N = 1k$ (55.8%), but degrades considerably to 44.6% when $N = 100$. This shows that the *cls* scores account for the accuracy of the highest ranked proposals.

On the other hand, when the *reg* layer is removed at test-time (so the proposals become anchor boxes), the mAP drops to 52.1%. This suggests that the high-quality proposals are mainly due to regressed positions. The anchor boxes alone are not sufficient for accurate detection.

Table 2: Detection results on **PASCAL VOC 2007 test set**. The detector is Fast R-CNN and VGG-16. Training data: "07": VOC 2007 trainval, "07+12": union set of VOC 2007 trainval and VOC 2012 trainval. For RPN, the train-time proposals for Fast R-CNN are 2k. †: this was reported in [5]; using the repository provided by this paper, this number is higher (68.0±0.3 in six runs).

| method | # proposals | data | mAP (%) | time (ms) |
|---|---|---|---|---|
| SS | 2k | 07 | 66.9† | 1830 |
| SS | 2k | 07+12 | 70.0 | 1830 |
| RPN+VGG, unshared | 300 | 07 | 68.5 | 342 |
| RPN+VGG, shared | 300 | 07 | 69.9 | **198** |
| RPN+VGG, shared | 300 | 07+12 | **73.2** | **198** |

Table 3: Detection results on **PASCAL VOC 2012 test set**. The detector is Fast R-CNN and VGG-16. Training data: "07": VOC 2007 trainval, "07++12": union set of VOC 2007 trainval+test and VOC 2012 trainval. For RPN, the train-time proposals for Fast R-CNN are 2k. †: http://host.robots.ox.ac.uk:8080/anonymous/HZJTQA.html. ‡: http://host.robots.ox.ac.uk:8080/anonymous/YNPLXB.html

| method | # proposals | data | mAP (%) |
|---|---|---|---|
| SS | 2k | 12 | 65.7 |
| SS | 2k | 07++12 | 68.4 |
| RPN+VGG, shared† | 300 | 12 | 67.0 |
| RPN+VGG, shared‡ | 300 | 07++12 | **70.4** |

Table 4: **Timing** (ms) on a K40 GPU, except SS proposal is evaluated in a CPU. "Region-wise" includes NMS, pooling, fc, and softmax. See our released code for the profiling of running time.

| model | system | conv | proposal | region-wise | total | rate |
|---|---|---|---|---|---|---|
| VGG | SS + Fast R-CNN | 146 | 1510 | 174 | 1830 | 0.5 fps |
| VGG | RPN + Fast R-CNN | 141 | **10** | 47 | **198** | **5 fps** |
| ZF | RPN + Fast R-CNN | 31 | **3** | 25 | **59** | **17 fps** |

We also evaluate the effects of more powerful networks on the proposal quality of RPN alone. We use VGG-16 to train the RPN, and still use the above detector of SS+ZF. The mAP improves from 56.8% (using RPN+ZF) to 59.2% (using RPN+VGG). This is a promising result, because it suggests that the proposal quality of RPN+VGG is better than that of RPN+ZF. Because proposals of RPN+ZF are competitive with SS (both are 58.7% when consistently used for training and testing), we may expect RPN+VGG to be better than SS. The following experiments justify this hypothesis.

**Detection Accuracy and Running Time of VGG-16.** Table 2 shows the results of VGG-16 for both proposal and detection. Using RPN+VGG, the Fast R-CNN result is 68.5% for *unshared* features, slightly higher than the SS baseline. As shown above, this is because the proposals generated by RPN+VGG are more accurate than SS. Unlike SS that is pre-defined, the RPN is actively trained and benefits from better networks. For the feature-*shared* variant, the result is 69.9%—better than the strong SS baseline, yet with nearly cost-free proposals. We further train the RPN and detection network on the union set of PASCAL VOC 2007 trainval and 2012 trainval, following [5]. The mAP is **73.2%**. On the PASCAL VOC 2012 test set (Table 3), our method has an mAP of **70.4%** trained on the union set of VOC 2007 trainval+test and VOC 2012 trainval, following [5].

In Table 4 we summarize the running time of the entire object detection system. SS takes 1-2 seconds depending on content (on average 1.51s), and Fast R-CNN with VGG-16 takes 320ms on 2k SS proposals (or 223ms if using SVD on fc layers [5]). Our system with VGG-16 takes in total **198ms** for both proposal and detection. With the conv features shared, the RPN alone only takes 10ms computing the additional layers. Our region-wise computation is also low, thanks to fewer proposals (300). Our system has a frame-rate of 17 fps with the ZF net.

**Analysis of Recall-to-IoU.** Next we compute the recall of proposals at different IoU ratios with ground-truth boxes. It is noteworthy that the Recall-to-IoU metric is just *loosely* [9, 8, 1] related to the ultimate detection accuracy. It is more appropriate to use this metric to *diagnose* the proposal method than to evaluate it.

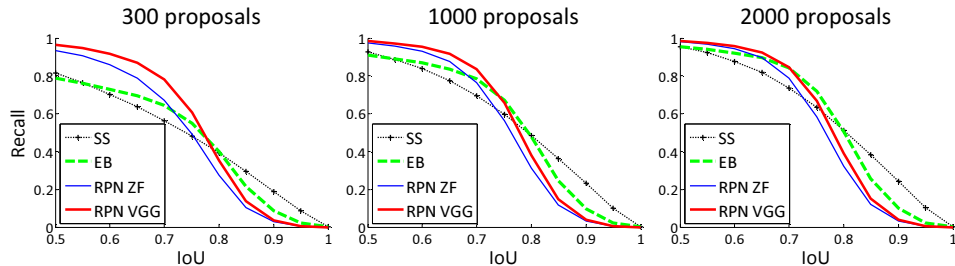

Figure 2: Recall *vs.* IoU overlap ratio on the PASCAL VOC 2007 test set.

Table 5: **One-Stage Detection *vs.* Two-Stage Proposal + Detection**. Detection results are on the PASCAL VOC 2007 test set using the ZF model and Fast R-CNN. RPN uses unshared features.

|  | regions | | detector | mAP (%) |
|---|---|---|---|---|
| Two-Stage | RPN + ZF, unshared | 300 | Fast R-CNN + ZF, 1 scale | 58.7 |
| One-Stage | dense, 3 scales, 3 asp. ratios | 20k | Fast R-CNN + ZF, 1 scale | 53.8 |
| One-Stage | dense, 3 scales, 3 asp. ratios | 20k | Fast R-CNN + ZF, 5 scales | 53.9 |

In Fig. 2, we show the results of using 300, 1k, and 2k proposals. We compare with SS and EB, and the $N$ proposals are the top-$N$ ranked ones based on the confidence generated by these methods. The plots show that the RPN method behaves gracefully when the number of proposals drops from 2k to 300. This explains why the RPN has a good ultimate detection mAP when using as few as 300 proposals. As we analyzed before, this property is mainly attributed to the *cls* term of the RPN. The recall of SS and EB drops more quickly than RPN when the proposals are fewer.

**One-Stage Detection *vs.* Two-Stage Proposal + Detection.** The OverFeat paper [18] proposes a detection method that uses regressors and classifiers on sliding windows over conv feature maps. OverFeat is a *one-stage*, *class-specific* detection pipeline, and ours is a *two-stage cascade* consisting of class-agnostic proposals and class-specific detections. In OverFeat, the region-wise features come from a sliding window of one aspect ratio over a scale pyramid. These features are used to simultaneously determine the location and category of objects. In RPN, the features are from square (3×3) sliding windows and predict proposals relative to anchors with different scales and aspect ratios. Though both methods use sliding windows, the region proposal task is only the first stage of RPN + Fast R-CNN—the detector *attends* to the proposals to refine them. In the second stage of our cascade, the region-wise features are adaptively pooled [7, 5] from proposal boxes that more faithfully cover the features of the regions. We believe these features lead to more accurate detections.

To compare the one-stage and two-stage systems, we *emulate* the OverFeat system (and thus also circumvent other differences of implementation details) by *one-stage* Fast R-CNN. In this system, the "proposals" are dense sliding windows of 3 scales (128, 256, 512) and 3 aspect ratios (1:1, 1:2, 2:1). Fast R-CNN is trained to predict class-specific scores and regress box locations from these sliding windows. Because the OverFeat system uses an image pyramid, we also evaluate using conv features extracted from 5 scales. We use those 5 scales as in [7, 5].

Table 5 compares the two-stage system and two variants of the one-stage system. Using the ZF model, the one-stage system has an mAP of 53.9%. This is lower than the two-stage system (58.7%) by 4.8%. This experiment justifies the effectiveness of cascaded region proposals and object detection. Similar observations are reported in [5, 13], where replacing SS region proposals with sliding windows leads to ∼6% degradation in both papers. We also note that the one-stage system is slower as it has considerably more proposals to process.

## 5   Conclusion

We have presented Region Proposal Networks (RPNs) for efficient and accurate region proposal generation. By sharing convolutional features with the down-stream detection network, the region proposal step is nearly cost-free. Our method enables a unified, deep-learning-based object detection system to run at 5-17 fps. The learned RPN also improves region proposal quality and thus the overall object detection accuracy.

## Footnotes

[1]"Region" is a generic term and in this paper we only consider *rectangular* regions, as is common for many methods (*e.g.*, [20, 22, 24]). "Objectness" measures membership to a set of object classes *vs.* background.

[2]For simplicity we implement the *cls* layer as a two-class softmax layer. Alternatively, one may use logistic regression to produce $k$ scores.

[3] In our early implementation (as also in the released code), $\lambda$ was set as 10, and the *cls* term in Eqn.(1) was normalized by the mini-batch size (*i.e.*, $N_{cls} = 256$) and the *reg* term was normalized by the number of anchor locations (*i.e.*, $N_{reg} \sim 2,400$). Both *cls* and *reg* terms are roughly equally weighted in this way.

[4] https://github.com/rbgirshick/fast-rcnn

[5]www.robots.ox.ac.uk/~vgg/research/very_deep/

[6]For RPN, the number of proposals (*e.g.*, 300) is the maximum number for an image. RPN may produce fewer proposals after NMS, and thus the average number of proposals is smaller.

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
