[Reviews · NeurIPS 2015]

Submitted by Assigned_Reviewer_1

The goal of the paper is clear and the experimental results are solid. However, the paper is not easy to follow with a few confusing parts mentioned below. 1. Why RPN is translation-invariant, but [19] is not. What about Overfeat?

2. Many parameters are set manually without further analysis. Just to name a few: Why n=3, k=9. 3. The training procedure seems complicated and not stable. Why not applying the same training procedure multiple times? Have the authors tried many other settings, but find the current one performs the best?
Summary: The authors propose a novel region proposal method which shares the CNN feature used for object localization. The region proposal method consists of a classification layer for recognizing object v.s. non-object, and a regression layer for predicting object bounding boxes. Due to sharing the feature computation with object localization module, the combined new RCNN is ~10x faster than fast RCNN and slightly more accurate. Since the training procedure of object localization CNN assumes regions are given, a multiple stages alternative training procedure is proposed.

Submitted by Assigned_Reviewer_2

The paper presents a region proposal network (RPN) which shares the conv features with R-CNN and includes two fully connected layers that predicts object bounds and objectness scores at each position. RPN is heuristically combined with Fast R-CNN for object detection and achieves state-of-the-art performance on VOC07 and VOC12. The running time of generating region proposals is greatly reduced since RPN shares the conv layers with the detection network.

Pros: - Smart idea on sharing conv layers between RPN and R-CNN which greatly reduced the run-time of object proposals.

- State-of-the-art performance on object detection.

- Well-written paper.

Cons:

- The training of RPN and fast R-CNN are weakly combined by alternating optimization and there is no convergence proof.

- RPN can be used independently as a region proposal method. However, it doesn't seem to outperform selective search.

Summary: This is a very nice piece of work, which improves both the efficiency and accuracy of R-CNN in object detection. Well-written paper and thorough experimental evaluations. The only concern is on how to combine RPN and fast R-CNN for object detection. The proposed alternating optimization method is a bit heuristic.

Submitted by Assigned_Reviewer_3

This work is a natural extension to the existing CNN based object detection. My concerns are as follows.

1. In the optimization section, it looks like there is only one iteration for the alternating optimization between RPN and R-CNN. If this is the case, it assumes the proposals generated by the first iterated RPN are optimal, which is obviously not true.

2. When the pixel-wise sliding window is applied to the top layer of CNN, what is the step size on the image layer? Since CNN has several rounds of max-pooling, the top level is in a much smaller scale compared with the input image. The sliding window on the top level may have a big step on the image level, causing inaccurate bounding box locations.

3. What is definition of $h_a$ in the line 173 and 174?

Summary: This work proposes a two stage object detection algorithm based on convolutional neural network (CNN). The first stage is region proposal, which is based on the traditional sliding window method but working on the top layer feature map of CNN (RPN). In the second stage, a fast R-CNN is applied to the proposed regions. Since the convolution layers are shared between RPN and R-CNN, and the calculation is speeded up using GPU, the algorithm can achieve near real-time (5fps).

Author Feedback
Author rebuttal: We sincerely thank all reviewers and AC for their time and effort.
Our paper proposes a method for learning object proposals as a ConvNet that shares features/computation with a Fast R-CNN detection network, leading to real-time detection rates with state-of-the-art accuracy. We are glad that the reviews are consistently positive, and we appreciate the constructive suggestions for improving our paper. The reviewers noted that "the paper will have real practical impact in object detection" (R4), "the experimental results are solid" (R1, R5), and the paper "will be very interesting to the computer vision community as the results are impressive" (R6).
We address the reviewer's concerns and questions as follows.

R1: "Why RPN is translation-invariant, but [19] is not. What about Overfeat?"
[19] (MultiBox) regresses 800 boxes from an entire image (implemented as a 224x224 input), and the 800 predictions are generated by a single *fully-connected* layer and therefore is not translation-invariant.
Overfeat is translation-invariant, but in a different manner. It is a "one-stage detection" system *without* proposals. A detailed discussion and comparison with the Overfeat's philosophy is in line 404-423.

R1: "Many parameters are set manually without further analysis ... Why n=3, k=9."
We agree that it will strengthen the paper if more investigations on parameters are provided. We will add this in a future extended version due to limited space.
The parameters in the paper are set as *simple* as possible, just for verifying the prototype idea:
n=3 (window size): this is the smallest nontrivial window size. Others are possible.
k=9 (# anchors): this is an outcome of 3 naive scales (128, 256, and 512) and 3 naive ratios (1:2, 1:1, and 2:1).
We believe that carefully tuning them might have better results on a particular dataset; but this is not our focus.

R1: "Why not applying the same training procedure multiple times? Have the authors tried many other settings, but find the current one performs the best?"
R3: "it looks like there is only one iteration for the alternating optimization between RPN and R-CNN"
We adopt the current procedure mainly because of concerns on training time, as each individual pass of RPN or Fast RCNN takes a half day training (for VGG). We did try running more iterations, but found the gain diminishing and not worthwhile in practice.

R2: "The proposed alternating optimization method is a bit heuristic"
We admitted (line 205-212) the current alternating optimization is a compromise. Our system is a 2-stage cascade of two networks, which is nontrivial for the current backprop method to optimize jointly. We believe joint optimization (of RPN + Fast RCNN) is interesting future work (line 211).

R2: "RPN can be used independently as a region proposal method. However, it doesn't seem to outperform selective search."
There are still debates on how to evaluate (independent) region proposal methods; see [9, 8, 1]. Fig. 2 shows that RPN behaves more gracefully than SS, when the number of proposals is small. When combined with a detector (Table 2), RPN is better than SS by 3% in mAP, suggesting that RPN is a better proposal method than SS. Also note that RPN is learned and can be more adaptive to the data, while SS is not learned.

R3: "what is the step size on the image layer? ... causing inaccurate bounding box locations"
The network step size (stride) is 16 pixels for ZF/VGG nets. Note that the input image is upsampled 2x (RPN/Fast R-CNN operate on ~1000x600 images, and original VOC images are ~500x300), so the effective stride is ~8 pixels. We will clarify.
We agree with R3 that there should be much room to improve accuracy, considering that such a big stride already works well.
There were methods investigating finer strides (see FCN paper [13]). But this is beyond the focus of this paper that proposes original, prototype ideas.

R3: "What is definition of $h_a$ in the line 173 and 174?"
h_a and w_a are height and width of the anchor boxes. We will clarify.